# Multivariate adjustment of drizzle bias using machine learning in European climate projections

Georgia Lazoglou[1], Theo Economou[1], Christina Anagnostopoulou[2], George Zittis[1], Anna Tzyrkalli[1], Pantelis Georgiades[1,3], and Jos Lelieveld[1,4]

[1]Climate and Atmosphere Research Centre (CARE-C), The Cyprus Institute, Nicosia, Cyprus
[2]Department of Meteorology and Climatology, School of Geology, Aristotle University of Thessaloniki, Thessaloniki, Greece
[3]Computation-based Science and Technology Research Center (CaSToRC), The Cyprus Institute, Nicosia, Cyprus
[4]Department of Atmospheric Chemistry, Max Planck Institute for Chemistry, Mainz, Germany

**Correspondence:** Georgia Lazoglou (g.lazoglou@cyi.ac.cy) and Jos Lelieveld (jos.lelieveld@mpic.de)

**Abstract.** Precipitation holds significant importance as a climate parameter in various applications, including studies on the impacts of climate change. However, its simulation or projection accuracy is low, primarily due to its high stochasticity. Specifically, climate models often overestimate the frequency of light rainy days while simultaneously underestimating the total amounts of extreme observed precipitation. This phenomenon, known as 'drizzle bias,' specifically refers to the model's tendency to overestimate the occurrence of light precipitation events. Consequently, even though the overall precipitation totals are generally well-represented, there is often a significant bias in the number of rainy days. The present study aims to minimize the "drizzle bias" in model output by developing and applying two statistical approaches. In the first approach, the number of rainy days is adjusted based on the assumption that the relationship between observed and simulated rainy days remains the same in time (thresholding). In the second, a machine learning method (Random Forests or RF) is used for the development of a statistical model that describes the relationship between several climate (modelled) variables and the observed number of wet days. The results demonstrate that employing a multivariate approach yields results that are comparable to the conventional thresholding approach when correcting sub-periods with similar climate characteristics. However, the importance of utilizing RF becomes evident when addressing periods exhibiting extreme events, marked by a significantly distinct frequency of rainy days. These disparities are particularly pronounced when considering higher temporal resolutions. Both methods are illustrated on data from three EURO-CORDEX climate models. The two approaches are trained during a calibration period and they are applied for the selected evaluation period.

## 1 introduction

Climate models are the fundamental tool for simulating historical conditions and projecting the future. However, due to the chaotic nature and the fine spatio-temporal scales of atmospheric processes, the ability to fully understand and model parts of the climate system is limited, resulting in incomplete representations of physical processes in climate models ( Maraun and Widmann (2018)). As a result, both global (GCM) and regional climate model (RCM) outputs tend to have systematic errors, commonly referred to as biases ( Jacob et al. (2007)). In general, biases tend to be more prominent for parameters characterized

by high stochasticity - such as precipitation ( Flato et al. (2014)). The process of minimizing these discrepancies is known as 'bias adjustment' or 'bias correction' (BC). This post-modeling procedure is of paramount importance, particularly for impact studies, as it significantly enhances the accuracy of results, including future climate projections ( Christensen et al. (2008)).

Precipitation is a key variable that has been extensively utilized in climate assessments and impact studies. However, due to its high volatility, climate models exhibit significant biases, both in terms of the total amount and its spatio-temporal distribution ( Goodison et al. (1998)). One prominent discrepancy that arises in the context of precipitation is in the simulated versus observed number of rainy days. The majority of climate models tend to overestimate the occurrence of rainy days with low precipitation, while simultaneously underestimating the intensity of more extreme events ( Maity et al. (2019)). This behaviour is widely recognized as the 'drizzle bias', where climate models tend to over-predict the occurrence of light precipitation events like drizzle (Argüeso et al. (2013); Gutowski Jr et al. (2003)). Consequently, while the overall precipitation totals are reasonably well represented, this is compensated by an excessive number of drizzle events. However, this notable disparity in the number of wet days affects several other precipitation statistics such as the standard deviation ( Baigorria et al. (2007)). Moreover, this phenomenon hinders accurate representation of temporal variability in climate models ( Maraun et al. (2017)).

Drizzle bias is of significant importance because it directly impacts decisions made in impact studies, such as those related to water resource management and agriculture. For instance, the presence of the drizzle bias hampers the accurate representation of precipitation attributes and the forecasting of hydrological extremes within climate models ( Trenberth et al. (2003)). Another noteworthy impact lies in the assessment of wet and dry spells, where long dry spells may wrongly manifest as shorter ones ( Maraun et al. (2017)). Agriculture is another domain significantly affected by drizzle bias, notably impacting the outcomes of crop models. Specifically, these models face challenges in accurately representing the water balance due to an excess of wet days, resulting in soil saturation before the onset of extreme events ( Dosio and Paruolo (2011)). This bias can introduce complexities in predicting and managing water resources within agricultural systems. Furthermore, the precision of simulations and forecasts of day-to-day precipitation is important in the reliability of forest fire predictions. Given the prominence of forest fires as a high-risk and multi-impact hazard, the discernible correlation between precipitation occurrence and the incidence of forest fires has been well-documented ( Argüeso et al. (2013)). Hence, precise rainfall representation is of critical significance in calculating fire indices, particularly those involving precipitation as an input (e.g., Fire Weather Index (FWI) Stocks et al. (1989)).

The majority of bias correction (BC) methods alter the least wet days to dry ones, redistributing precipitation amounts over the remaining wet days. In this way, the drizzle phenomenon is addressed simultaneously with the correction of the rainfall amount. A recent study by Pan et al. (2021) attempts to address the frequency of rainy days as part of their bias correction methodology by applying an adversarial learning method. This data-driven approach improves the representation of precipitation frequency and intensity. Generally, more sophisticated methods such as machine learning approaches have shown significant potential to enhance the accuracy of precipitation values. This assertion is supported by another recent study Hess et al. (2022) where the authors utilized an approach based on neural networks to improve the local distribution and spatial structure of precipitation simulations while maintaining low computational costs. In this context, the correction of both the number of dry days and the intensity of precipitation was achieved simultaneously. Additionally, Fulton et al. (2023) employed

an approach based on neural networks, specifically the Unsupervised Image-to-Image Translation (UNIT) neural network, to bias-correct climate parameters. They combined this with other simpler bias correction methods such as quantile mapping to increase the accuracy of the results. In their study, the drizzle phenomenon is corrected simultaneously with the correction of precipitation values, while temperature is also included to allow for dynamic improvement. As such, a comprehensive treatment of both the frequency of dry days and the intensity of precipitation is achieved.

The concept of splitting the BC of precipitation into two steps - first addressing the correction of dry-day frequency and then the wet-day intensity - is a promising approach Smitha et al. (2018)). Such a two-step approach for bias correction of precipitation is also discussed by Pierce et al. (2015), with an effort to preserve the daily precipitation variability using a ratio or percentage change factor. This shows potential for enhancing the accuracy of precipitation modeling but it also has limitations in locations that are dry and have insufficient precipitation days.

To the best of our knowledge, there is a dearth of studies thoroughly investigating this topic, hence it is a distinct shortcoming that must be addressed before adjusting actual precipitation amounts. Furthermore, it is widely acknowledged that a more in-depth analysis of this subject is necessary, along with a better understanding of how multivariate methods impact the structure of time series( Van de Velde et al. (2020)).

A study on this two-step approach was conducted by  Van de Velde et al. (2020), who divided the precipitation BC process into two stages: correcting precipitation occurrence (the number of rainy days) and precipitation intensity (the amount of precipitation). Three univariate methods for correcting rainy days were tested, and the results were then combined using both univariate and multivariate approaches to adjust rainfall amounts. The study concluded that the simplest method, 'thresholding,' yields better results when compared to other methods, even though they may have a higher level of complexity. Thresholding has also been used in several other studies where it was important to correct the frequency of wet days. For instance, Ines and Hansen (2006) used thresholding for correcting the mean monthly frequency of rainy data to make them more suitable as inputs to crop models. Schmidli et al. (2006) tried to improve the accuracy of simulated precipitation values, by removing, separately, the bias in wet-day frequency and intensity, using the general idea of the thresholding method. However, other studies emphasize the advantage of more complex univariate approaches ( Vrac et al. (2016)).

Overall, it is widely acknowledged that the correction process becomes unstable when there is a significant disparity between the observed and simulated frequency of rainy days, particularly for methods assuming temporal stationarity in the correction ( Switanek et al. (2017)). Some recent methods have emerged, building upon the concept of Quantile Mapping, such as the Scaled Distribution Mapping method (SDM) ( Switanek et al. (2017)).

Recently, extensive research has been conducted to address the BC of precipitation, particularly at higher temporal reso-lutions ( Lazoglou et al. (2020)). However, not many researchers are focusing on the challenging matter of drizzle BC. This issue plays a pivotal role in the broader context of correcting biases in daily precipitation amounts. In this study, we address the drizzle bias issue using both univariate and multivariate approaches. We aim to determine the optimal statistical approach for enhancing the accuracy of simulated and projected rainy-day counts in the broader Euro-Mediterranean region.

## 2 Data and Methodology

### 2.1 Data

This study utilizes daily observations collected over 25 years, from 1981 to 2005. The data was sourced from the Global Summary of the Day (GSOD), provided by the National Climatic Data Center (NCDC) ( GSOD (2022)). GSOD offers comprehensive coverage of daily meteorological measurements from ground stations. These data underwent rigorous quality control procedures to eliminate random errors. To ensure the robustness of the analysis, we only considered stations with less than 5% of missing values. Figure 1 depicts the location of the 600 stations included in the analysis.

Additionally, for a more focused analysis, the studied area has been split into specific sub-regions which have similar characteristics. Based on the PRUDENCE project (Prediction of Regional Scenarios and Uncertainties for Defining European Climate Change Risks and Effects) which was a research initiative that aimed to assess the regional climate change impacts over Europe, we divided Europe into sub-domains for more detailed analysis (Figure 1) ( Christensen and Christensen (2007)). Namely, the 10 PRUDENCE regions are the Iberian Peninsula (IP), France (FR), Mid-Europe (ME), the Alps (AL), the Mediterranean (MD), 6 Eastern Europe (EA), Northwestern Africa (NA), the Middle East (MI), Scandinavia (SC), and the British Isles (BI). Stations that are not included in any of these sub-domains are excluded from this part of the analysis.

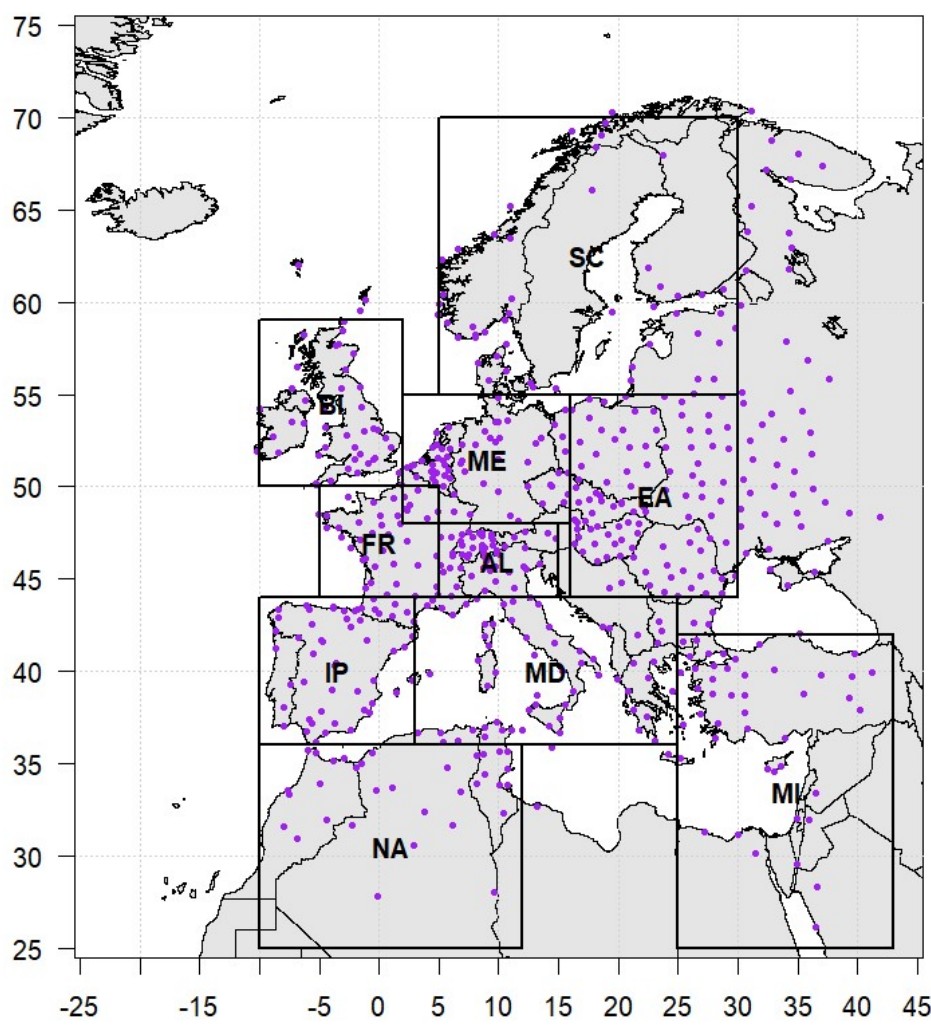

**Figure 1.** Location of the 600 available stations for the period 1981-2005 and definition of sub-domains: 1. Iberian Peninsula (IP), 2. France (FR), 3. Mid-Europe (ME), 4. Alps (AL), 5. Mediterranean (MD), 6. Eastern Europe (EA), 7. Northwestern Africa (NA), 8. Middle East (MI), 9. Scandinavia (SC), 10. British Isles (BI).

In conjunction with ground station data, this study incorporates daily fields from three EURO-CORDEX climate models (Jacob et al., 2020), serving as the target input data for correction (Table 1). The three climate models considered were chosen to evaluate BC methods across various regional climate models (RCMs) with distinct initial conditions from different global climate models (GCMs). We selected models exhibiting different behaviour in key climate parameters such as precipitation,

temperature, and the frequency of projected wet days (including the drizzle phenomenon), which is the primary focus of this study. Given that many climate models tend to overestimate the number of dry days, our emphasis was on addressing this. The boxplots in Figure 2 illustrate for each RCM the total annual precipitation, the number of rainy days, and the mean annual temperature for the grid cells closest to the 600 GSOD stations, over the 1981-2005 period. In line with common practice for distinguishing between rainy and non-rainy days using a threshold of 0.1mm, we have adopted this criterion for the present research ( Anagnostopoulou and Tolika (2012); Liu et al. (2013); Lehner et al. (2020)). Another commonly used threshold is 1mm (e.g. Velasquez et al. (2020)), however given that the present study includes areas such as the Mediterranean where the phenomenon of drought is crucial, we consider the 0.1mm threshold to be more appropriate. On average, the first RCM is the driest and the coldest one, whereas its rainy-day average number aligns closely with the other two models. The second RCM is the wettest, while the third one simulates the highest temperatures. Notably, the disparity in rainy days between the second and third models is marginal, except for differences observed in extreme conditions. The RCM output considered in our analysis includes daily precipitation, mean temperature, and relative and specific humidity. The nearest continental grid to each station was selected for applying the BC.

**Table 1.** List of assessed EURO-CORDEX Regional Climate Models (RCMs).

| Model | Driving Global Model | Regional Climate Model |
|-------|---------------------|------------------------|
| RCM1 | CNRM-CERFACS-CNRM-CM5 | KNMI-RACMO22E |
| RCM2 | MPI-M-MPI-ESM-LR | SMHI-RCA4 |
| RCM3 | NCC-NorESM1-M | GERICS-REMO2015 |

## 2.2 Methodology

The objective of this study is to assess the efficacy of univariate and multivariate BC methods in addressing the 'Drizzle Bias' in climate models. The investigation utilizes daily time series of precipitation collected from 600 stations across the broader Euro-Mediterranean region, as well as timeseries of various climate parameters from three EURO-CORDEX climate models. Specifically, the study evaluates the performance of two distinct methods—'Thresholding' and 'Random Forests'—in enhancing the precision of modelled rainy day occurrences. The two methods are assessed and compared using both 'standard' and 'extreme deviation' cases. To accomplish this, a systematic approach has been adopted, involving a series of steps implemented for each of the 600 stations. The whole procedure is applied separately to each grid point, meaning that each station is treated as a unique case, and the behaviour of the rest does not affect the others. Due to this, information for the coordinates of the stations is not needed in the analysis. The steps that have been followed are described below and are depicted in Figure 3.

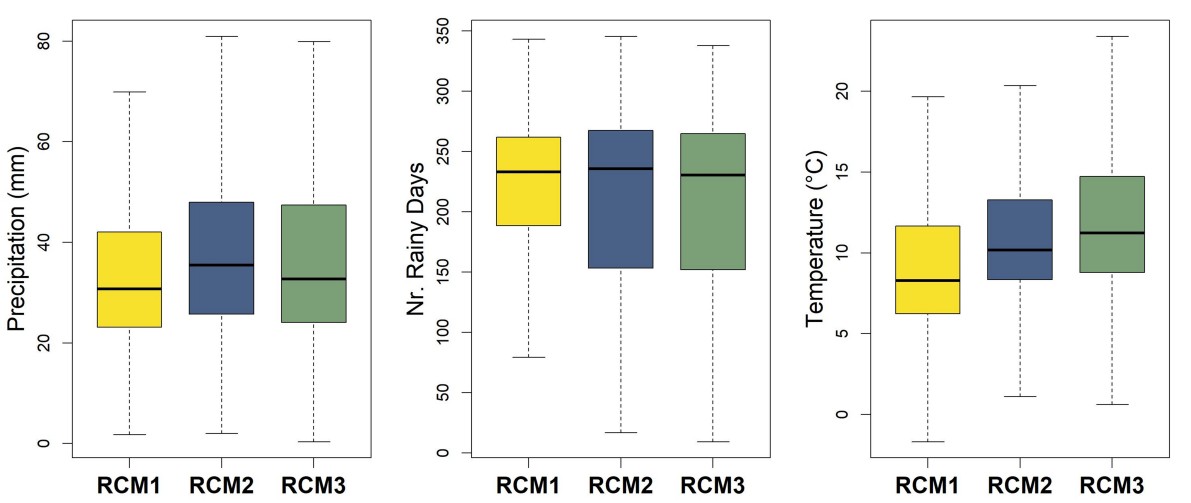

**Figure 2.** Comparison of the three EURO-CORDEX models based on their annual precipitation total, number of rainy days and mean annual temperature, averaged for the period 1981-2005.

**Data Splitting**

**Standard Cases**
- Calibration Period (1981-2000)
- Evaluation Period (2001-2005)

**Extreme Deviation Cases**
- Calibration Period (Non-extreme years)
- Evaluation Period (Extreme deviation years)

**Monthly Statistics Computation**
- Observations (*Rainy Days*)
- Simulations (*Rainy Days, Total Precipitation, Mean Temperature, Mean Specific Humidity, Mean Relative Humidity*)

**Model Training**

**Thresholding**
Observed and simulated counts of rainy days

**Random Forest**
All available parameters

**Model Application**
Apply trained methods to data from the evaluation period

**Evaluation**
Compare results with observed data in the evaluation period

**Figure 3.** Methodology Steps Flowchart

1. The initial stage of the methodology includes segregating observed and simulated time series data into two distinct periods: calibration and evaluation. This separation facilitates the identification of both 'standard' and 'extreme deviation' cases.

   - In the 'standard case', the evaluation period directly follows the calibration period. In our 25-year dataset, the initial 20 years (1981-2000) constitute the calibration period. The subsequent five years (2001-2005) serve as the target years for BC and are used for method evaluation.

   - The term 'extreme deviation case' refers to target years for which the discrepancy between the observed and simulated number of rainy days is extreme. Specifically, for each station and each year between 1981 and 2005, the difference between the simulated and observed rainy days frequency is calculated. Subsequently, the 95th percentile of these differences is computed. Years exhibiting differences surpassing this threshold are identified as extreme deviation cases. These selected years constitute the target data for BC evaluation, while the remaining years serve as the calibration period. The target dataset for this analysis typically comprises 2-3 years.

2. After the definition of the calibration and evaluation periods, monthly statistics of all climate parameters are computed. For the observations, the total count of rainy days is computed. For the simulated data we estimate the sum of rainy days, total precipitation, mean temperature, as well as mean specific and relative humidity.

3. After establishing the final monthly database, the selected BC methods ('Thresholding' and 'Random Forest') are applied to model the relationship between the observed count of wet days and the simulated data. The 'Thresholding' method uses only the observed and simulated counts of rainy days, while the 'Random Forest' incorporates additional climate model output variables. Both BC methods are trained using data from the calibration period, assuming that the BC mechanism remains the same over time. Subsequently, each trained method is applied to data from the evaluation period, to correct the number of rainy days. The choice of the thresholding approach was guided by the aim to employ a univariate, simple, and well-established BC method that yields satisfactory outcomes. This method was selected as the 'baseline' for comparative analysis alongside a more sophisticated multivariate method, which offers the flexibility to incorporate several additional parameters easily. The two methods are described in what follows.

   **Thresholding** is a widely used method for correcting the frequency of wet days - 'occurrence bias adjusting method' ( Van de Velde et al. (2020)). It is an effective and easy-to-use method, but mostly applied in cases where the model simulates more wet days than the observations ( Vrac et al. (2016)). In its basic form, thresholding involves converting all simulated values below a designated threshold to zero. The more refined implementation aims to equate the count of days falling below this threshold in the simulated data to the corresponding count in the observed data. This study adopts the thresholding approach detailed by  Van de Velde et al. (2020). Initially, dry day frequencies in both observed and simulated datasets are computed. Subsequently, the difference between these frequencies defined the associated count of days in the simulated data that required adjustment. The simulated wet days are ordered based on ascending precipitation amounts and the appropriate number of days with the lowest precipitation are set to zero, in order to make

the two frequencies match. Importantly, these adjustments are performed on a monthly basis to preserve a realistic temporal structure.

**Random Forest** is a tree-based method, known for its flexibility and robustness, and has been proven to be a powerful ensemble learning algorithm for predictive modelling and machine learning tasks. The algorithm constructs an array of decision trees during training and combines their predictions to improve accuracy and generalisation. In the Random

Forest framework, each decision tree is constructed using a random subset of the features (here climate model output), which effectively reduces over-fitting and enhances the model's ability to capture complex relationships in the data ( Breiman (2001a)). Finally, the output prediction is obtained by aggregating the predictions of all trees. In the Random Forest Regressor, as used in this study, the output prediction is obtained by averaging the decision trees. This ensemble approach provides a powerful prediction tool but also offers resilience to outliers and noisy data ( Hastie et al. (2009)).

For the training and evaluation procedures, the Random Forest Regressor implementation included in the randomForest (version 4.7-1.1) package in R programming language was utilised (Breiman, 2001b). The specific package was used for both feature selection and model training. Initially, feature importance was assessed using the 'random.forest.importance' function, which calculates the importance of each predictor variable. The feature set used to train the models comprised the following model-generated variables: total number of rainy days in each month, monthly precipitation sum, monthly mean near-surface

relative humidity, monthly mean near-surface specific humidity, monthly mean near-surface air temperature, and month of the year. These variables were utilized to build the random forest model in the calibration period, while the target variable for the random forest model was the total number of rainy days in the evaluation period. These steps have been followed both for the 'Standard Cases' as well as for the 'Extreme Deviation Cases'. We employed a total number of 2500 trees in the forest and set the importance parameter to TRUE, to compute variable importance scores for feature selection. Additionally, other parameters

such as node size were left at their default values to maintain model stability. Lastly, it is important to mention that the whole procedure was followed separately for each station. Hence each station is treated as a unique case and a different random forest model is calculated using all the available variables for the whole calibration period.

## 3   Results

### 3.1   RCM Drizzle Bias in Europe and the Mediterranean

To assess the extent of drizzle bias, we initiate by contrasting the observed frequency of rainy days with the simulated occurrences generated by three widely used RCMs. The analysis is performed both for the default evaluation period (2001-2005) (standard case) and the extreme deviation cases. Figure 4 illustrates the annual frequency of rainy days based on station observations (left panels) and the corresponding model biases, for the standard 5-year and extreme deviation evaluation periods (right panels). The results relate to one of the three climate models, however, similar behaviour is evident in the other models

(Supplementary Material - Figures Sup1 and Sup2).

Figure 4 (top panels) indicates a consistent latitudinal change in the number of rainy days but also the degree of bias, varying from the northern to the southern regions. Stations in northern Europe experience over 260 rainy days annually, in sharp contrast with the Mediterranean region, where the number is generally below 80 days per year. Notably, the southernmost stations record the lowest number at around 20 wet days per year.

In terms of the bias, Figure 4 indicates that across the entire area, the model consistently overestimates rainy days, as indicated by universally positive differences. In most stations, the percentage overestimation is around 80%. However, there are many exceptions, notably in the Mediterranean area, where the overestimation surpasses 100% due to lower rainy day counts. Moreover, the scale of overestimation reveals the existence of stations where the model simulates over 200% more relative to observed wet days. These stations are detected in northwest Africa, the Alpine region, and near the Black Sea. Similar patterns were found in the analysis of the other two models (Supplementary Material - Figures Sup1 and Sup2). Notably, the second model yields higher overestimation, particularly in southern areas, where percentiles frequently exceed 150%.

The bottom row of Figure 4 (second-row maps) presents the extreme deviation cases, with different outcomes than the standard case (top row). Northwest Europe records higher rainy day numbers, with high frequencies in the eastern part also showing substantial counts (> 120 days). Conversely, the Balkan Peninsula reports fewer wet days, while the Mediterranean consistently has the lowest counts. Notably, during these extreme deviation years, stations in Africa did not exceed 25 rainy days annually. These maps (Figure 4) originate from years displaying extreme behaviour, posing considerable challenges for climate models. This complexity is evident in the bottom right map of Figure 4, where percentiles indicate an overestimation of recorded wet days by the initial model surpassing 100% across all stations. In areas with the highest rainy day frequency, overestimation hovers around 150%. Notably, a significant number of stations exhibit deviations exceeding 200%, covering vast regions like the Mediterranean, Poland, and the Balkans. This pattern aligns with findings from the other two climate models.

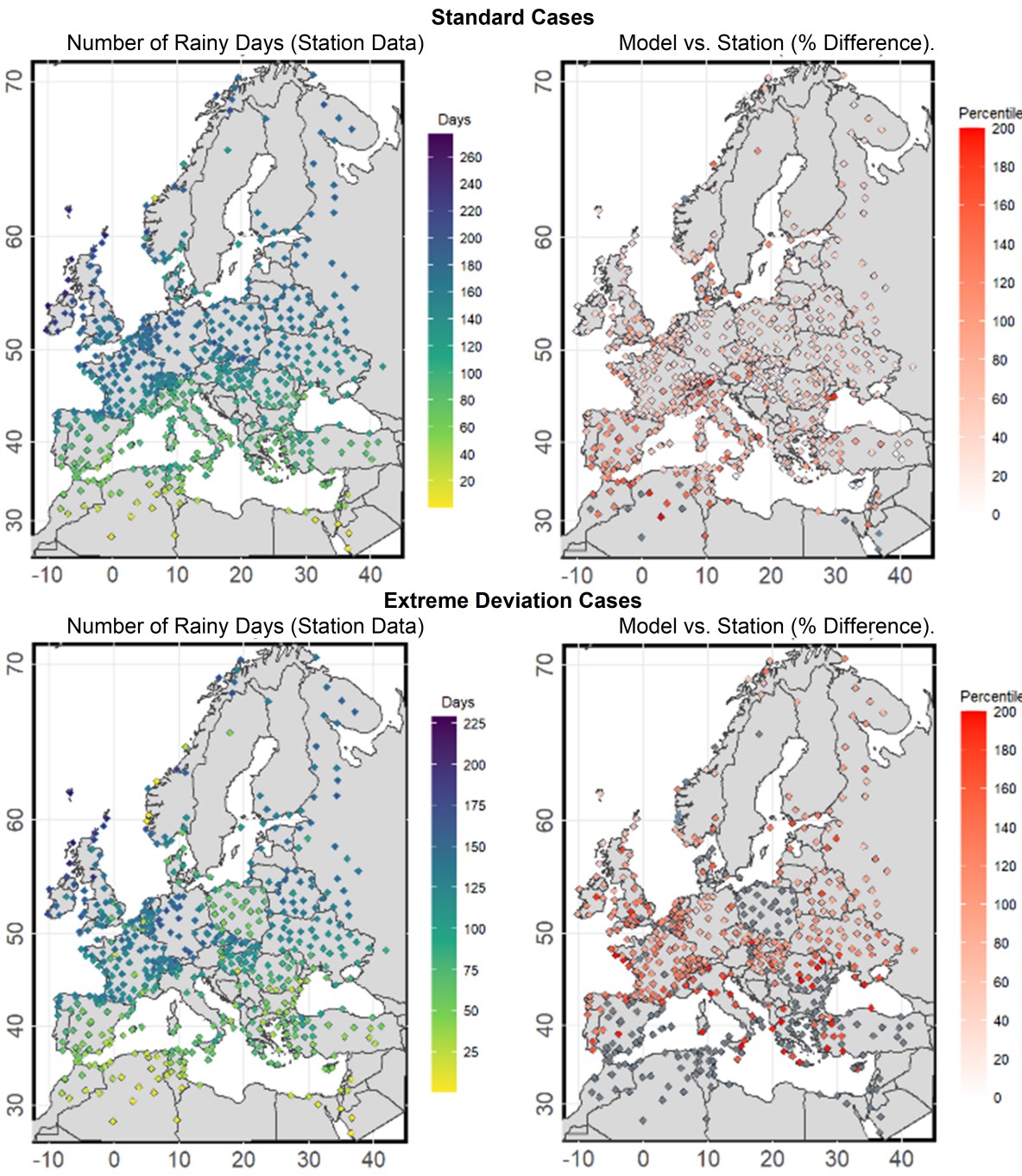

**Figure 4.** Mean annual frequency of rainy days for the Standard and Extreme Deviation cases (left panel) and the bias of rainy days (in percentage) based on the first model (right panel). Locations with biases greater than 200% are highlighted in grey.

## 3.2 Method Comparison: Station Improvement Percentiles

The relative performance of the two methods is summarised in Table 2, in terms of the percentage of stations (600 in total) that either approach 'wins', or indeed if there is equal performance. The quantity used to measure performance is the difference between the bias corrected number of rainy days and the observed number of rainy days. The results are shown for both monthly and annual time scales, for each of the three climate models and for both the standard and extreme deviation cases. The relative performance results are also shown for a subset of stations, where the difference in performance between the two methods is significant. Here, 'significance' is defined as instances where the variance in the number of corrected rainy days between methods exceeds 5% of the annual rainy days for each station. For instance, consider a scenario where the initial count of rainy days at a station is 100 days. Hence, the 5% of the rainy days of this station is equal to 5 days. Employing the Rainfall Factor (RF) method yielded a corrected count of 105 days, whereas the Thresholding method resulted in 107 days. The difference between the corrected values derived from these two distinct methods amounts to 2 days. This difference of 2 days falls below the 5-day threshold we established for this station. Consequently, this discrepancy fails to meet the established threshold for significance, indicating that the variance between the two correction methodologies is not statistically meaningful within the confines of this station's characteristics.

This detailed investigation allows for a focused analysis of the accuracy and effectiveness of the two methods. In the subsequent in-depth analysis (column diff > 5%), this comparison includes stations where the corrected rainy days from the two methods are not precisely equal but remain very close—less than 5% bias from the observations. This nuanced comparison, in the focused analysis, specifically focuses on differences exceeding the aforementioned 5% threshold in corrected rainy day counts.

At an annual scale and for the standard cases (evaluation period 2001-2005), the percentage of stations exhibiting equal performance between the two methods remains consistently below 2% for all models. However, for the 'diff $< 5\%$' case (termed here case 1 as opposed to case 0), these percentages significantly rise in all models, surpassing 70%. For case 0 the RF method consistently yields greater accuracy. For models RCM1 and RCM2 the annual count of rainy days is closer to observations compared to the adjustments derived from the thresholding method in 65% of the 600 stations. However, for case 1, the thresholding method achieves a slight advantage for RCM1 and RCM2, with percentages of 4–8% greater than those of the RF, as sometimes the complexity of the variables adds some noise. Nonetheless, for RCM3, a different trend emerges as RF prevails in both case 0 and case 1.

The clear superiority of the RF method, which incorporates various climate parameters, becomes more pronounced in the extreme deviation cases. On a yearly basis, the percentage of stations showing equal performance of the two methods is effectively zero for all three models for case 0, while for case 1 this raised to about 20–30%. It is clear however the RF method is overwhelmingly better for the extreme deviation cases, for both case 0 and case 1, for all three models. In that particular, the RF method is superior for about 90% of the stations across all three models. The corresponding percentages for case 1 exceed 62%.

**Table 2.** Percentage of stations for which either method performs best, or equal performance at both Yearly and Monthly temporal scales, and for both standard and extreme cases. The percentages are computed for all stations (case 0), but also for the subset of stations where the difference in performance is greater than 5% (case 1). 'Equal' means that the results from thresholding and RF are equal; 'Thresholding' means that the bias corrected results from thresholding are closer to the station compared to the RF while the 'RF' implies the reverse situation. For the case 1 (differnece $> 5\%$), 'Equal' means the difference is $\leq 5\%$.

| | | Yearly | | | | Monthly | | | |
| | | Standard Cases | | Extreme Deviation Cases | | Standard Cases | | Extreme Deviation Cases | |
| Model | Performance | case 0 | case 1 | case 0 | case 1 | case 0 | case 1 | case 0 | case 1 |
|---|---|---|---|---|---|---|---|---|---|
| | Equal | 0 | 76 | 0 | 33 | 5 | 38 | 1 | 20 |
| RCM1 | Thresholding | 37 | 14 | 11 | 5 | 25 | 24 | 27 | 30 |
| | RF | 63 | 10 | 88 | 62 | 70 | 37 | 72 | 50 |
| | Equal | 1 | 70 | 2 | 18 | 8 | 35 | 1 | 19 |
| RCM2 | Thresholding | 32 | 18 | 10 | 7 | 24 | 23 | 24 | 27 |
| | RF | 67 | 10 | 90 | 74 | 70 | 42 | 74 | 54 |
| | Equal | 2 | 72 | 0 | 23 | 8 | 34 | 2 | 20 |
| RCM3 | Thresholding | 43 | 9 | 8 | 5 | 23 | 23 | 25 | 28 |
| | RF | 56 | 19 | 92 | 72 | 70 | 42 | 73 | 52 |

In the monthly analysis, the results reflect the superiority of the RF method in correcting the number of rainy days compared to the thresholding method. This is the case for both the standard and extreme deviation cases and is consistent for all models. Specifically, under the standard case, the percentage of stations with the two methods having an equal performance ranges from 5 to 8%, in case 0. This number increases to around 35% when considering case 1. Looking at the standard case and case 0 highlights that in 70% of the stations, the RF corrections prove to be more accurate. Moreover, for case 1, the RF method maintains substantially higher percentages compared to thresholding.

The monthly analysis for the extreme deviation cases indicates that stations exhibiting equal results between the methods constitute less than 2% of the total for case 0, while for case 1 the number rises to about 20%. Then, for both case 0 and case 1, the RF method significantly outperforms the thresholding method, with percentages of $> 70\%$ for case 0 and $> 50\%$ for case 1.

### 3.3 Direct Spatial Comparison of the two BC Methods

Figure 5 provides a visual comparison between the two BC methods. The absolute difference between the observations and the bias corrected data (for each method) is computed for the annual count of rainy days. The difference between these differences then defines the relative accuracy of the two methods. Positive values (shown in red) indicate a better performance of the

thresholding method, while negative values (blue) denote the opposite. The analysis is shown for RCM1, with corresponding
figures for the other two models presented in the supplementary material (Figures Sup3 and Sup4).

The left panel of Figure 5 illustrates that, for the standard cases, both methods exhibit comparable performance across most stations. The differences between the two approaches range from -4 to 4 days in most of the stations (white colour). A prevalence of the thresholding method is mainly obtained for Eastern Europe and a few locations in the Mediterranean region. However, for stations with large differences between the methods, the RF method demonstrates better performance.
This is particularly evident in the Balkan Peninsula, where the differences are on average, approximately 16 rainy days per year. Additionally, the RF method is superior in other regions across central Europe. The right panel of Figure 5 (extreme deviation cases), demonstrates a pronounced dominance of the RF method. Negative values are universally observed across the studied area. In central Europe, disparities range from 2 to 12 days, with higher ranges evident in other regions. Notably, in the Mediterranean region and its eastern sectors, differences exceed 25 days in certain stations. These variations are also recorded
in parts of central and northeastern Europe.

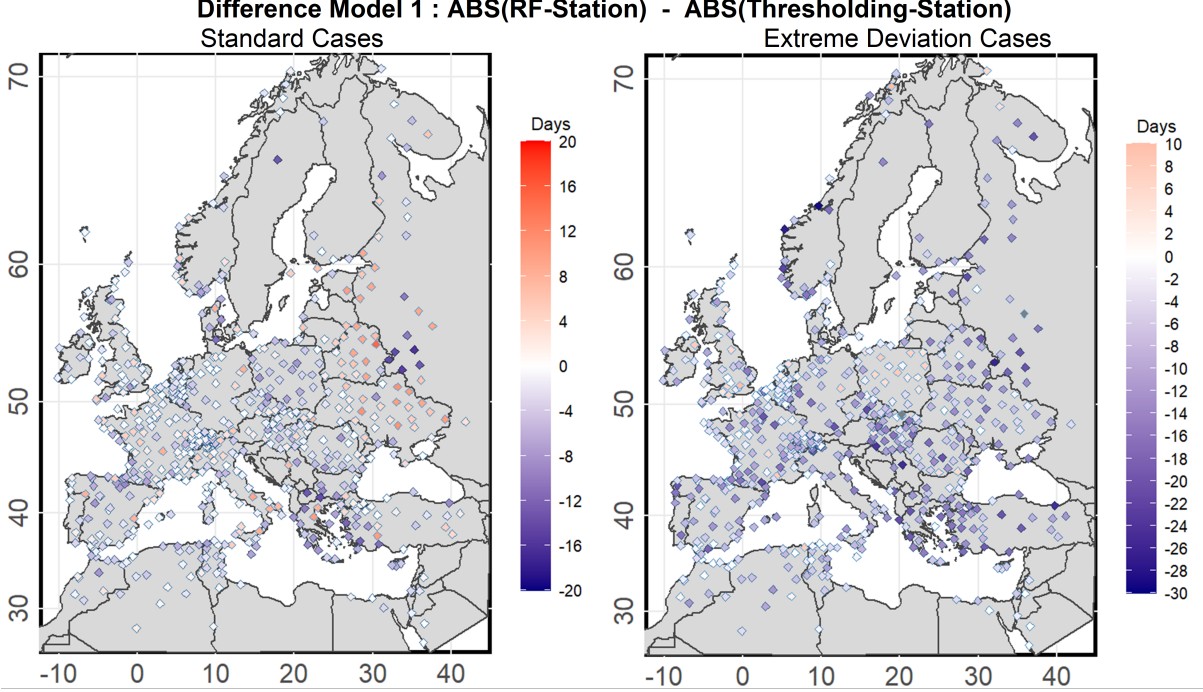

**Figure 5.** Comparison of Absolute Deviations: absolute value of RF Method minus Observed values vs. absolute value of Thresholding Method minus Observed values for Annual Corrected Rainy Days

Overall, the results from all three models indicate that both methods offer similar accuracy in correcting simulated rainy days across the majority of stations during normal periods. However, the RF method exhibits better skill in some regions relative

to the thresholding method. In all three models, this disparity is amplified for the extreme deviation cases, highlighting the superiority of the RF method across the studied area.

## 3.4 Focused analysis on European subdomains

To better understand the relative performance of the two methods across different climatic conditions, we utilise Quantile-Quantile (Q-Q) plots across the 10 sub-areas defined earlier. These plots facilitate a graphical comparison of two probability distributions by plotting their quantiles against each other. Figures 6 and 7 show Q-Q plots of the annual number of rainy days in the standard and extreme deviation cases respectively. The results from the other two models exhibit similar behaviour and are provided in the supplementary material (Figures Sup5 - Sup8). In each figure, there are two groups of panels. In the first group (top two rows), the raw model output and its corresponding bias-corrected values from each method are compared to quantiles from the observations. In the second group (bottom two rows), the raw model output is omitted to facilitate a more clear analysis of the two BC methods.

The first group of Q-Q plots in Figure 6 demonstrates that, in standard cases, both thresholding and RF methods contribute to enhancing the accuracy of simulated rainy day numbers across all sub-domains. Notably, the model consistently tends to overestimate observed numbers, as indicated by the green line being above the diagonal line across all areas, unlike the bias corrected quantiles. A more focused comparison between the two methods shown in the second group of Q-Q plots reveals the advantage of the RF over the thresholding method in multiple areas. Specifically, in 'SC,' 'BI,' 'FR,', 'ME' and 'MI' areas, the thresholding line diverges from the diagonal, particularly in the upper or lower tails. This suggests that for the standard cases, the inclusion of other climate parameters can significantly aid in correcting the tails of the count distribution. In these areas, the RF line better aligns with the diagonal line representing the default observed dataset with high accuracy. In the remaining areas, both methods yield very good and nearly identical results.

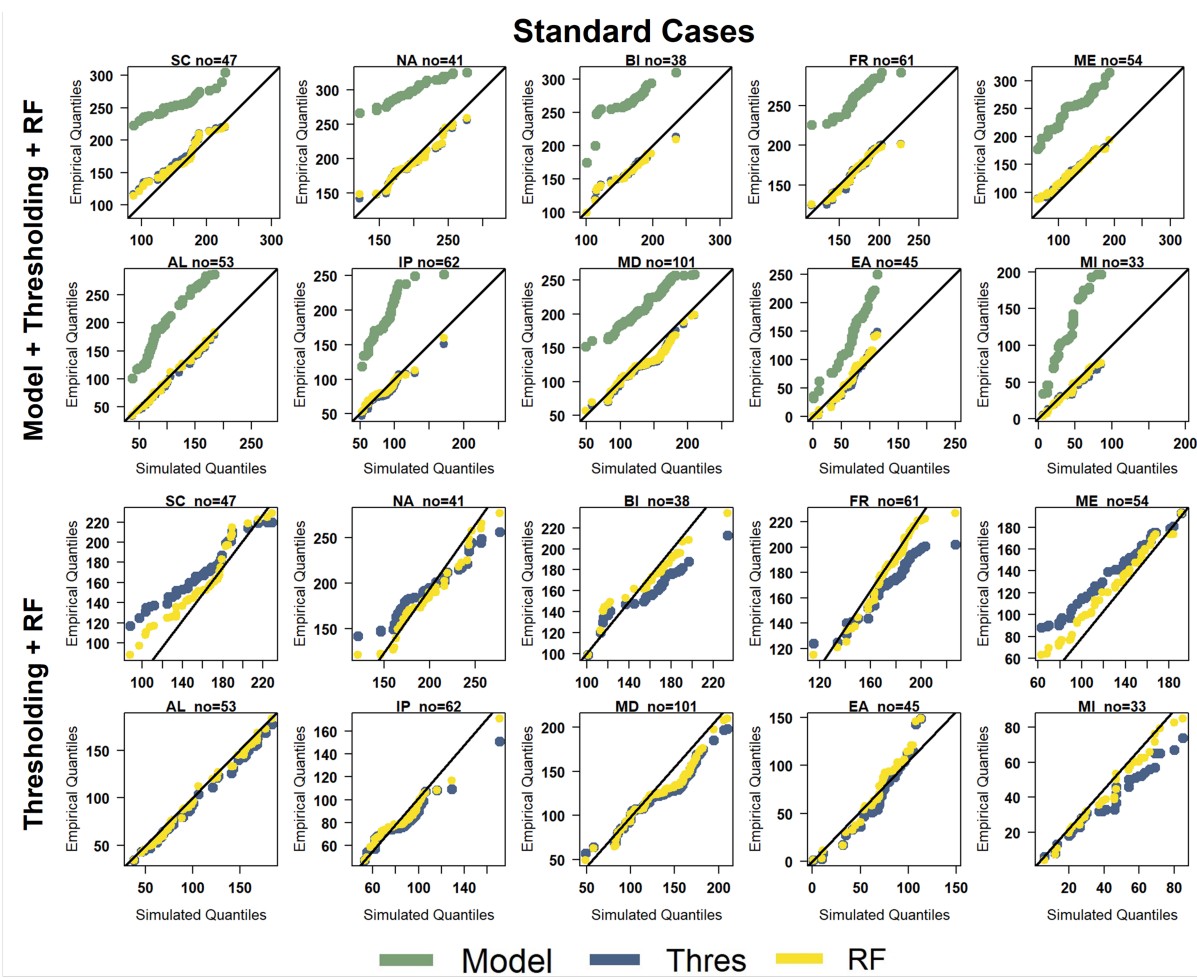

**Figure 6.** Q-Q Plots Comparing the number of rainy days (per year) for the default evaluation period (2001-2005): Simulated values from the first model (green) and their respective corrected values derived from both the thresholding method (blue) and the RF method (yellow) are depicted. The first set of Q-Q plots displays outcomes for the 10 sub-areas, encompassing the model values. The second set illustrates the identical outcomes, specifically focusing on the BC methods.

The significance of this general outcome becomes more pronounced in the extreme deviation cases shown in Figure 7. The model predicts significantly more annual rainy days across all sub-areas, with some areas showing simulated values three times higher than the recorded ones. Consequently, both methods effectively correct these discrepancies. However, upon closer inspection of the two BC methods, a distinct advantage emerges for RF. Across seven of the 10 sub-areas, the RF lines more closely align with the diagonal, while the divergence of the thresholding line, particularly in the tails, is notably significant. In the remaining three areas ('AL,' 'EA,' 'MI'), the adjustments made by both methods are remarkably similar.

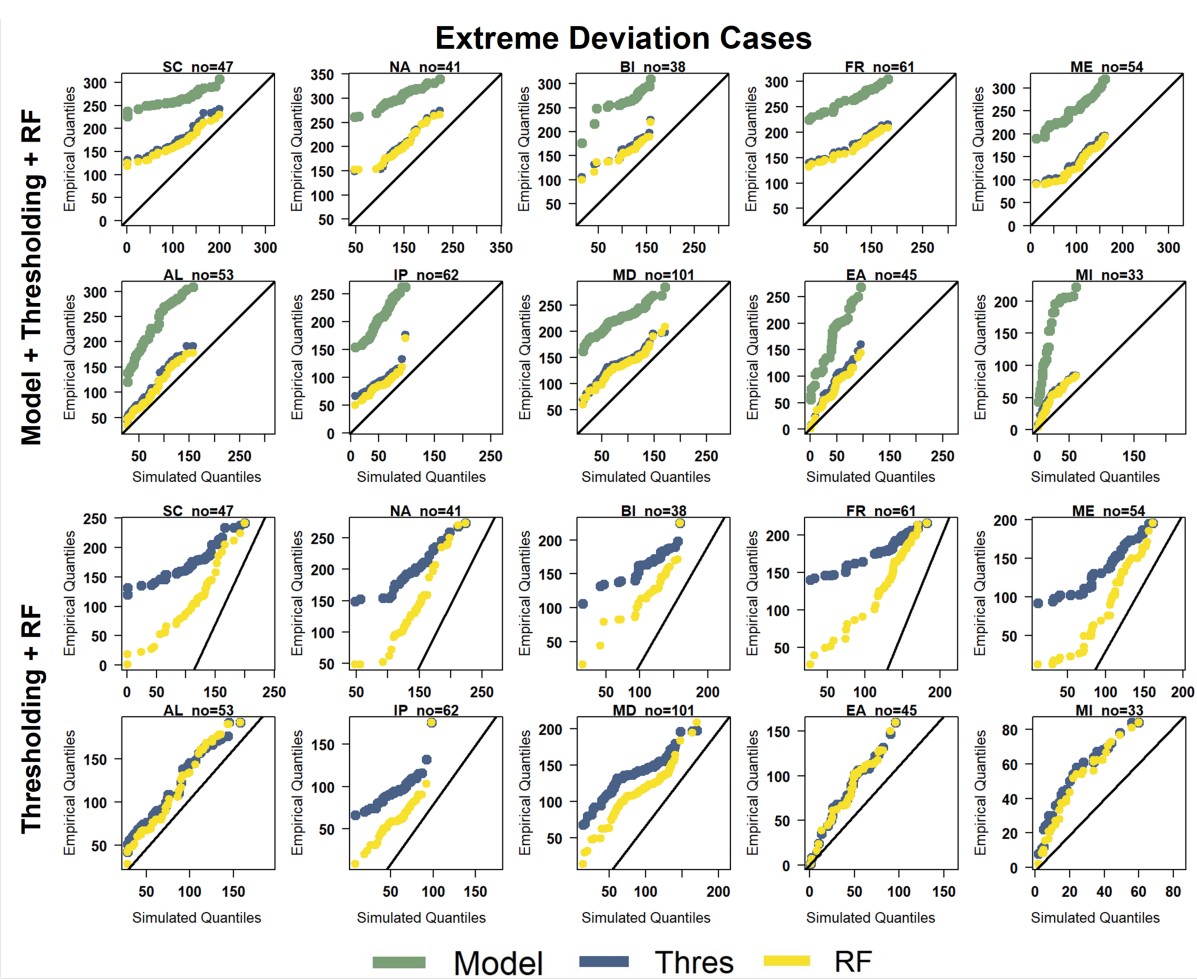

**Figure 7.** QQ Plots Comparing the Number of Rainy Days in Extreme Deviation Cases: Simulated values from the first model (green) and their respective corrected values derived from both the thresholding method (blue) and the RF method (yellow) are depicted. The first set of QQ plots displays outcomes for the 10 sub-areas, encompassing the model values. The second set illustrates the identical outcomes, specifically focusing on the BC methods.

## 3.5 Degree of Improvement for the European subdomains

Besides the visual analysis conducted for the 10 sub-domains, Table 3 offers a percentage-based examination of improvement facilitated by the two BC methods. This table shows corresponding percentages as in Table 2 for both the standard and extreme deviation cases and all three models. The numbers relate to the annual count of rainy days.

Table 3 indicates the superiority of the RF for RCM1 across almost all sub-areas. Notably, in 'SC,' 'IP,' 'EA,' 'MI,' and 'AL,' the percentages relating to the RF method are nearly double relative to thresholding. Conversely, in the 'BI,' 'FR,' and 'MD' 310 areas, the results are more evenly distributed, with each method covering nearly half of the stations. Outcomes from extreme

deviation cases reveal a markedly different picture. In all areas, station percentages relating to the RF exceed 70%, with 5 out of 10 areas reaching a percentage exceeding 90%.

For RCM2, the results are similar in the extreme deviation cases, where the RF is superior in over 80% of the stations within each area. For the standard deviation cases, the RF is superior for most areas although differences are small when thresholding has the advantage.

For RCM3, the RF is again overwhelmingly superior of for the extreme deviation cases. The results are a bit moremore volatile for the standard cases, where the Thresholding method exhibits higher accuracy in the 'FR' and 'NA' sub-areas, comparable accuracy in the 'MD' and 'ME' sub-areas, while across the remaining six sub-areas, the RF method dominates with numbers of over 60%.

**Table 3.** Percentages of improvement of the three models, for the 10 sub-areas on a Yearly scale, considering Standard and Extreme Deviation cases.

| | RCM1 | | | | RCM2 | | | | RCM3 | | | |
|---|---|---|---|---|---|---|---|---|---|---|---|---|
| | Standard Cases | | Extreme Deviation Cases | | Standard Cases | | Extreme Deviation Cases | | Standard Cases | | Extreme Deviation Cases | |
| | Thres-holding | RF | Thres-holding | RF | Thres-holding | RF | Thres-holding | RF | Thres-holding | RF | Thres-holding | RF |
| SC | 31 | 69 | 4 | 96 | 54 | 44 | 10 | 90 | 27 | 73 | 8 | 92 |
| BI | 44 | 56 | 29 | 71 | 24 | 76 | 15 | 85 | 41 | 59 | 7 | 93 |
| FR | 55 | 45 | 21 | 79 | 29 | 71 | 5 | 95 | 68 | 32 | 13 | 87 |
| ME | 38 | 62 | 8 | 92 | 16 | 84 | 10 | 90 | 46 | 52 | 7 | 93 |
| AL | 27 | 73 | 18 | 82 | 45 | 55 | 5 | 95 | 24 | 75 | 11 | 89 |
| IP | 32 | 68 | 2 | 98 | 57 | 43 | 4 | 96 | 40 | 60 | 11 | 89 |
| MD | 44 | 56 | 6 | 94 | 29 | 71 | 6 | 94 | 40 | 60 | 3 | 97 |
| EA | 29 | 71 | 18 | 82 | 21 | 78 | 21 | 79 | 52 | 47 | 10 | 90 |
| MI | 31 | 69 | 0 | 100 | 31 | 64 | 4 | 93 | 38 | 62 | 9 | 91 |
| NA | 73 | 27 | 9 | 91 | 24 | 76 | 0 | 100 | 88 | 12 | 0 | 100 |

## 4 Discussion and Conclusions

This study focused on the important issue of the drizzle bias effect in regional climate models, described by an over-prediction of the number of rainy days while underestimating associated precipitation amounts. The primary objective was to ascertain an optimal statistical approach aimed at enhancing the accuracy of projected rainy days in the broader Euro-Mediterranean region. To this end, two distinct methodologies were applied and rigorously evaluated. The first method known as 'Thresholding' is a type of change-factor approach, which assumes that the proportion of the difference between observed and simulated rainy

days remains the same. The second method is the 'Random Forest', a machine learning multivariate approach that encompasses various climate parameters.

The analysis of data from three climate models highlights a significant drizzle bias issue in the studied area, both on monthly and yearly time scales, for standard and extreme deviation cases. This finding is consistent with prior studies emphasizing the significance of the drizzle bias effect. For instance, Maraun et al. (2017) highlights the challenge in modeling temporal variability beyond the drizzle effect, which substantially impacts various analyses, including the duration of wet and dry spells. Additionally, the drizzle phenomenon affects the model's physical parameterizations by depleting moisture essential to represent extreme rainfall events ( Jerez et al. (2013), Gianotti et al. (2012)). This effect becomes more pronounced during years of extreme deviation behaviour, as confirmed by our results. There have been attempts to overcome the drizzle based on a wet day threshold (1 mm) leading to overcorrections, affecting also the representation of extremes ( Maraun (2013)). The general outcome of this study is the need for bias correction, showing the advantages which per the present research clarifies the advantages of a more complex multivariate method.

The superiority of the multivariate method, exemplified in this research by Random Forest (RF), stems from its ability to incorporate multiple climate parameters. A further analysis with respect to variable importance was conducred, using of the SHapley Additive exPlanations (SHAP) plot which is a model-agnostic tool derived from game theory that assigns each feature an importance value for a particular prediction (Lundberg and Lee, 2017). The associated SHAP file (supplementary material) revealed that mean temperature holds particular importance in correcting the drizzle bias, in that high temperatures negatively impact the prediction of rainy days. Furthermore, specific humidity emerges as the second most significant climate parameter influencing the bias, with lower humidity levels positively contributing to predictions. Thus, the inclusion of at least these two parameters in the multivariate approach leads to more accurate results. Nonetheless, a more focused investigation into the varying impacts of several climate parameters on the drizzle phenomenon is recommended.

Moreover, the intensity of the drizzle phenomenon varies significantly across specific cases or regions. In areas with monsoon seasons, the 'drizzle effect' is notably intensified, at times doubling or tripling compared to observations ( Smitha et al. (2018)). Similar overestimation, particularly in extreme deviation cases, have been found for various European regions, corroborating our results. Specifically, our study identifies these areas predominantly within the Mediterranean domain, where the driest stations are located. This finding aligns with the notion that these regions experience significant rainfall intermittence, thereby amplifying the drizzle effect, as previously noted by Ines and Hansen (2006). In addition, Olsson et al. (2016) mentioned the intensity of dry bias is the frequency of precipitation in the southern part of Europe during summer. Our spatial analysis demonstrates the methodological accuracy employed in this study, revealing the broad enhancements seen across the entire region with the application of RF methods, which are similarly reflected in areas sharing analogous characteristics. Notably, in the Middle East (MI) and the Iberian Peninsula (IP), RF methods significantly outperform thresholding for extreme cases across almost all stations. Furthermore, in standard cases, better performance of RF is obtained for approximately 70% of subareas within these regions.

Upon clarifying the phenomenon's significance in the studied area, we applied two methods to minimize the drizzle bias. The findings reveal that, for the standard cases and the yearly temporal scale, while the RF method notably enhances accuracy

across most stations, differences between the two approaches are generally insignificant. These discrepancies amount to less than 5% of the total rainy days per station. Employing a multivariate approach yields outcomes comparable to the default threshold method in correcting temporal coherence within analogous climate periods. However, at a higher temporal resolution - monthly scale - the RF method demonstrates superiority. Notably, the utility of Random Forest (RF) becomes apparent when dealing with periods characterized by extreme deviation behaviour, featuring markedly different frequencies of rainy days. In these instances, the RF method outperforms thresholding, especially at higher temporal resolutions.

These results are encouraging for using the RF method, as the thresholding method is one of the most efficient univariate methods for occurrence-bias-correction of precipitation ( Van de Velde et al. (2020)). In particular, Van de Velde et al. (2020) compared the thresholding method with two univariate methods, 'Stochastic Singularity Removal (SSR)' ( Vrac et al. (2016)) and 'Triangular Distribution Adjustment (TDA)' ( Pham et al. (2018)), to test its ability to adjust the frequency of rainy days for the area of Belgium. The results proved that the randomness included in the SSR and TDA methods, performed generally worse than thresholding. However, this differs from Vrac et al. Vrac et al. (2016), who highlighted SSR's superiority over other methods, although the overall performance among the univariate methods was similar. To the best of the authors' knowledge, there have been no similar endeavours in using multivariate methods for occurrence bias correction. However, the effectiveness of multivariate methods has been proved in correcting precipitation intensity biases (e.g. Piani and Haerter (2012)), and maybe their efficiency might stem from their capability to consider inter-variable, spatial, and temporal properties ( Vrac and Friederichs (2015)).

In summary, this study significantly contributes to the field by addressing the crucial correction of the number of wet days—an essential preliminary step in correcting daily precipitation biases for future climate scenarios. In particular, using the information for the number of drizzle days, the days with the lowest rainfall amounts are counted as dry days. It should be mentioned that as the present study is conducted on a monthly basis, one cannot specify which particular days of the month were dry – just their total number. Our findings indicate that for 'standard' years, both the widely used univariate thresholding method and the multivariate machine learning approach of Random Forest (RF) demonstrate comparable accuracy in addressing drizzle bias on a yearly basis. However, as temporal resolution increases, the predominance of the RF method becomes more pronounced. Our results strongly suggest that employing the RF method is highly advisable, particularly when dealing with target years for bias correction that may exhibit extreme behaviour. The RF method proves to be considerably more accurate than the univariate method, chiefly due to its incorporation of several other climate parameters. Nevertheless, further comprehensive analyses and studies are warranted to fully assess the broader implications and potential applications of these findings.

*Code and data availability.* All the data, code and supplementary material are available and can be accessed at Zenodo with DOI: 10.5281/zenodo.10468125

*Author contributions.* G.L. and T.E. conceived and designed the project. A.T. and G.L. performed data curation and quality control. G.L., T.E. and P.G. performed calculations and data analysis. G.L. and T.E. led manuscript writing. G.L., T.E., C.A., G.Z and J.L. interpreted the results and provided general scientific input, critical review and overall support. All authors assisted in manuscript writing and preparation.

*Competing interests.* The authors declare that they have no conflict of interest.

*Acknowledgements.* This research was supported by the PREVENT project that has received funding from the European Union's Horizon Europe Research and Innovation Program under Grant Agreement No. 101081276. It was also supported by the EMME-CARE project that has received funding from the European Union's Horizon 2020 Research and Innovation Program, under Grant Agreement No. 856612, as well as matching co-funding by the Government of Cyprus.

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
