# Peer review of "Multivariate adjustment of drizzle bias using machine learning in European climate projections"

_EGUsphere, 2024_

## Referee Comment (RC2)

This is a generally well-written manuscript focusing on the adjustment of "drizzle bias". The topic is interesting as the climate models often overestimate the frequency of light rainy days while simultaneously underestimating the total amounts of extreme observed precipitation. Two statistical approaches were used in this research, including the thresholding and the Random Forest.

However, there is considerable room for improvement in this paper.

1. Paragraph 50, introduction, "The majority of bias correction (BC) methods alter the least wet days to dry, redistributing precipitation amounts over the remaining wet days"
I understand that the author wants to emphasize the importance drizzle bias correction, but there are many bias correction methods that do not follow the two-step method at present, but directly correct all precipitation days. For example, based on deep learning: cycle Gan (Pan et al., 2021; Hess, et al., 2022), UNIT (Fulton et al., 2023), etc., traditional bias correction: QM, CDF-t (Pierce et al., 2015), etc. These methods need to be mentioned if they have shortage on drizzle bias correction.

Pan B, Anderson G J, Goncalves A, et al. Learning to correct climate projection biases. Journal of Advances in Modeling Earth Systems, 2021, 13(10): e2021MS002509.
Hess P, Drüke M, Petri S, et al. Physically constrained generative adversarial networks for improving precipitation fields from Earth system models. Nature Machine Intelligence, 2022, 4(10): 828-839.
Fulton D J, Clarke B J, Hegerl G C. Bias correcting climate model simulations using unpaired image-to-image translation networks. Artificial Intelligence for the Earth Systems, 2023, 2(2): e220031.
Pierce D W, Cayan D R, Maurer E P, et al. Improved bias correction techniques for hydrological simulations of climate change. Journal of Hydrometeorology, 2015, 16(6): 2421-2442.

2. Paragraph 90. "The selection of these models was deliberate, aimed at encompassing a range of model configurations" This sentence is ambiguous. Please give more reasonable reasons why you chose these three models. Maybe, there are obvious differences in the precipitation simulation results of these three models. They can represent different results of overestimate and underestimate of precipitation simulations.

3. As the threshold of rainy day is important, why you chose 0.1mm as an example?

4. Paragraph 145. The RF model structure should be more clear. 1) Establish a uniform model for all the month, or for each month, you generate a different model? 2) More detailed on model structure should be added, how many training data and test dataset; parameter setting such as depth or leaf node number, etc.

5. It will be clear if the author adds a flowchart figure to show the steps of the method part.

6. Paragraph 155. How to set month of the year as an input feature? Set March as 3? The model can be applied in all of the grid, or for each grid you generate a separate model? I notice there is no location (latitude and longitude) information in the training feature, how the model can identify different grid point if it is a uniform model can be applied on all of the grids?

7. For figure 3 and 4, it will be more clear if the regional division box in Figure 1 is placed on these maps, readers will be able to better correspond to the regions mentioned in the text.

8. Paragraph 340 " an essential preliminary step in correcting daily precipitation biases for future climate scenarios." After getting the ratio or the number of drizzle day, would the model help to tell which days these drizzle days will occur?

---

## Author Response (AR1)

CARE-C, The Cyprus Institute
Konstantinou Kavafi 20, Aglantzia
Nicosia, 2121
Cyprus

Email: g.lazoglou@cyi.ac.cy

27th April 2024

Dear editor,

Re: "**Multivariate adjustment of drizzle bias using machine learning in European climate projections**"

Thank you and the reviewers for taking the time to review our paper in such detail. We appreciate the chance to revise this paper.

Below is a point-by-point response detailing how we have addressed the suggestions (responses are in green). We feel that many comments and suggestions have strengthened the paper, in terms of clarity, robustness of the results, reproducibility and utility of the findings.

In addition, we have also performed a thorough reading of the manuscript and have fixed a number of typos and grammatical errors.

We look forward to hearing from you further at your convenience.

Yours sincerely,

Georgia Lazoglou and co-authors

**Reviewer 1**

This paper does a good job of taking the reader through bias corrections for three regional climate models which overestimate the number of rainy days and underestimate the precipitation amounts compared to annual and monthly observations. They demonstate that random forest regression outperforms thresholding for station comparisons where the model-measurement difference exceeds 5%. The figures and discussion are clear, and I have only minor comments:

**Response**: Thank you for taking the time to review our paper.

1) I had hoped for some insight into why the random forest regression succeeded when thresholding failed. Perhaps a paragraph or two in the discussion could cover next steps for learning which of the 5 feature set variables mattered most, and how to use that information for model improvement?

**Response**: We agree, and have now conducted further analysis with respect to the importance of the climate parameters that in the RF. Specifically, a SHAP (SHapley Additive exPlanations) diagram (a model-agnostic tool derived from game theory that assigns each feature an importance value for a particular prediction) was produced and included in the supplementary material, while a relevant paragraph (line 345) was added to the discussion. In short, mean temperature has considerable importance in correcting the drizzle bias, i.e. high temperatures negatively impact the prediction of rainy days. Furthermore, specific humidity emerges as the second most significant climate parameter influencing this phenomenon, with lower humidity levels positively contributing to predictions. The diagram is also given here for convenience.

[Figure]

2)  On line 152 the authors state that they used scikit-learn for the random forest model, but as far as I can see the zenodo archive contains only R code?

**Response**:  Unfortunately, this was a mistake on our part. All analyses were conducted using R, although we also used Python to double-check the results – which is how the Python details ended up in the paper. All results of the paper can be easily reproduced by using the R code available in the Zenodo file and we have now amended the text to only mention the R implementation in line 180.

3) The captions for Figures 5 and 6 say that the model and rf data are red and orange, but in the figures, they are green and yellow

**Response**: The captions have now been changed (new figures 6 and 7).

4) For the Q-Q plot in figure 5, standard case 48 thresholding does especially poorly -- any idea why this gridcell is an outlier?

**Response**: After some more analysis, we found that for the particular case  thresholding has poor performance because of the data in one station. This station is the small island of Ona in Norway (lon = 62.867, lat = 6.533) in which the observed number of rainy days is very low. However, we believe this is a spurious case because the measurements came from an automatic weather station and the values do not fit the climatic conditions at all. Therefore, we assume this discrepancy is due to a wrong measurement and have consequently excluded this station from all analyses.

5) the column headers for Table 2 and Table 3 use different labels for the same quantities -- case 0, case 1, "standard cases", "extreme deviation case" "div > 5%" - why not just adopt the unambiguous labels of Table 3?

**Response**: Firstly, as the reviewer correctly noticed in table 2 there were some labels like "dif> 5%" which were wrong. We have now fixed the titles in Table 2 to be equal in all columns. However, as table 3 is presenting the results for the 10 subdomains, we have done the comparison between the two methods (thresholding and RF) without getting deeper in the cases when the difference is lower than 5%. So, we analyzed case 0 for the 3 models and the 10 subdomains. That is the reason why we kept the main titles: Standard Cases and Extreme Deviation Cases (as in table 2) but not the subtitles case 0 and case 1. Instead of having these as secondary titles we have now included the two methods in columns.

6) it is stated that GridSearchCV was used to establish "the optimal set of hyper-parameters". It would be useful to state what the hyper-parameters were, along with the feature set variables

**Response**: Again we must apologize, this relates to your comment number 2. The initial mention of GridSearchCV and hyper-parameters indeed stemmed from our preliminary analysis conducted in Python. However, we acknowledge our oversight in not clarifying the shift to R and the randomForest package for the final analysis. We have now updated the specific paragraph to provide a detailed explanation of how we utilized the randomForest package in R, including the variables used in the feature set and the hyperparameter set for the model.

Chief editor comment:

I would like to know a minor issue in your manuscript. Currently, it contains a "Data availability" section and a "Code and data availability section". This is not according to the format of the manuscripts in our journal. You can have a "Code and Data Availability" section in your manuscript with the information for both types of asset, or as recommended in the guidelines for manuscripts, a "Code availability" and a "Data availability" section. Then, all the code should be mentioned in the first one, and the data in the second one.

Please, correct this issue in any reviewed version of your manuscript.

**Response**: We have now addressed this by incorporating a single "Code and Data Availability" section in our manuscript. In this section, we provide the Zenodo DOI address, where both the utilized data and the codes employed for generating the results of the manuscript are included within a designated folder.

**Reviewer 2**

This is a generally well-written manuscript focusing on the adjustment of "drizzle bias". The topic is interesting as the climate models often overestimate the frequency of light rainy days while simultaneously underestimating the total amounts of extreme observed precipitation. Two statistical approaches were used in this research, including the thresholding and the Random Forest.

However, there is considerable room for improvement in this paper.

**Response**: Thank you.

1. Paragraph 50, introduction, "The majority of bias correction (BC) methods alter the least wet days to dry, redistributing precipitation amounts over the remaining wet days". I understand that the author wants to emphasize the importance of drizzle bias correction, but there are many bias correction methods that do not follow the two-step method at present, but directly correct all precipitation days.

For example, based on deep learning: cycle Gan (Pan et al., 2021; Hess, et al., 2022), UNIT (Fulton et al., 2023), etc., traditional bias correction: QM, CDF-t (Pierce et al., 2015), etc. These methods need to be mentioned if they have shortage on drizzle bias correction.

1. Pan B, Anderson G J, Goncalves A, et al. Learning to correct climate projection biases. Journal of Advances in Modeling Earth Systems, 2021, 13(10): e2021MS002509.
2. Hess P, Drüke M, Petri S, et al. Physically constrained generative adversarial networks for improving precipitation fields from Earth system models. Nature Machine Intelligence, 2022, 4(10): 828-839.
3. Fulton D J, Clarke B J, Hegerl G C. Bias correcting climate model simulations using unpaired image-to-image translation networks. Artificial Intelligence for the Earth Systems, 2023, 2(2): e220031.
4. Pierce D W, Cayan D R, Maurer E P, et al. Improved bias correction techniques for hydrological simulations of climate change. Journal of Hydrometeorology, 2015, 16(6): 2421-2442.

**Response**: We agree and so the proposed references have been all added in the introduction (paragraphs 50 and 65).

2. Paragraph 90. "The selection of these models was deliberate, aimed at encompassing a range of model configurations" This sentence is ambiguous. Please give more

reasonable reasons why you chose these three models. Maybe, there are obvious differences in the precipitation simulation results of these three models. They can represent different results of overestimate and underestimate of precipitation simulations.

**Response**: We have now changed this sentence (now paragraph 110) to one that is much more clear.

3. As the threshold of rainy day is important, why you chose 0.1mm as an example?

**Response**: The choice of 0.1mm as the threshold for rainy days is generally accepted and frequently used in meteorological research to distinguish between wet and dry days. We understand that the reviewer's question arises from our omission to explicitly mention this in the manuscript. To address this, we have added an extension to the aforementioned sentence, explaining that this threshold is commonly proposed for precipitation classification, along with references supporting its usage in the field (paragraph 115).

4. Paragraph 145. The RF model structure should be more clear. 1) Establish a uniform model for all the month, or for each month, you generate a different model? 2) More detailed on model structure should be added, how many training data and test dataset; parameter setting such as depth or leaf node number, etc.

**Response**: We have now taken steps to enhance the clarity of the random forest model structure. In the revised paragraph detailing the procedures for training and evaluating the random forest models, we have provided comprehensive information on the model development process using the randomForest package in R. Specifically, for the model structure, we utilized the randomForest function with a total of 2500 trees in the forest. The importance parameter was set to TRUE to compute variable importance scores for feature selection. Other parameters such as nodesize were maintained at their default values to ensure model stability. Furthermore, to provide additional clarity on the model structure, we have included details on the training and evaluation datasets used. Last but not least we are mentioning that each station is treated separately and different random forest model are calculated using all the available variables for the whole calibration period (paragraph 180-190).

5. It will be clear if the author adds a flowchart figure to show the steps of the method part.

**Response**: We agree and have now added a flow chart was added in the manuscript. (Figure 3)

6. Paragraph 155. How to set month of the year as an input feature? Set March as 3? The model can be applied in all of the grid, or for each grid you generate a separate model? I notice there is no location (latitude and longitude) information in the training feature, how the model can identify different grid point if it is a uniform model can be applied on all of the grids?

**Response**: We understand that the upcoming comment arises from our omission to mention all these details in the text. Hence, a detailed explanation has been added in the part of methodology (paragraph 130).

The way that the month of the year is included in the model is exactly as the reviewer mentioned, numbering the months. This information has been added to paragraph 185.

7. For figure 3 and 4, it will be more clear if the regional division box in Figure 1 is placed on these maps, readers will be able to better correspond to the regions mentioned in the text.

**Response**: We understand the point, however we believe it is more suitable for the structure of the paper to maintain the maps without the inclusion of subarea boxes in Figures 3 and 4. Our rationale behind this decision lies in the focus of these figures, which primarily emphasize the overall assessment of the entire area, particularly concerning the total number of stations and their improvements.

In contrast, the detailed analysis of subareas is comprehensively covered in the manuscript's dedicated section, "3.4. Focused Analysis on European Subdomains," where such assessments are presented through boxplots and tables. By maintaining this approach, we aim to ensure clarity and coherence in the structure of our paper, thus minimizing any potential confusion for our readers.

8. Paragraph 340 "an essential preliminary step in correcting daily precipitation biases for future climate scenarios." After getting the ratio or the number of drizzle day, would the model help to tell which days these drizzle days will occur?

**Response**:  No, this is not currently possible with the particular model setup, although this is something we are working on as one of the next steps of this work. We have now added a new paragraph to reflect this (paragraph 385).